REGISTERED REPORT PROTOCOL

# The bio-sonographic index. A novel modality for early detection of acute kidney injury after complex vascular surgery. A protocol for an exploratory prospective study

**Ahmed Zaky** [1]*, **Adam W. Beck** [2], **Sejong Bae** [3], **Adam Sturdivant** [1], **Amandiy Liwo** [1], **Novak Zdenek** [2], **Nicole McAnally** [2], **Shama Ahmad** [1], **Brad Meers** [1], **Michelle Robbin** [4], **J. F. Pittet** [1], **Ashita Tolwani** [5], **Dan Berkowitz** [1]

1 Department of Anesthesiology and Perioperative Medicine, University of Alabama at Birmingham, Birmingham, Alabama, United States of America, 2 Department of Surgery, Division of Vascular Surgery and Endovascular Therapy, University of Alabama at Birmingham, Birmingham, Alabama, United States of America, 3 Department of Medicine, Division of Preventive Medicine, University of Alabama at Birmingham, Birmingham, Alabama, United States of America, 4 Department of Radiology, University of Alabama at Birmingham, Birmingham, Alabama, United States of America, 5 Department of Medicine, Division of Nephrology, University of Alabama at Birmingham, Birmingham, Alabama, United States of America

* azaky@uabmc.edu

This is a Registered Report and may have an associated publication; please check the article page on the journal site for any related articles.

## Abstract

### Objective

Acute kidney injury (AKI) is a common complication of complex aortic surgery with high mortality, morbidity and health care expense. The current definition of AKI does not allow for structural characterization of the kidneys and utilizes functional indices with substantial limitations leading to delayed diagnosis and ineffective interventions. The aim of this study is to develop a method of early detection of structural renal abnormalities that can precede and predict the occurrence of AKI in this population. We propose a novel combined index of ultrasonography (shear wave elastography), biomarkers of renal stress (urinary insulin growth factor binding protein-7, IGFBP-7 and inhibitor of tissue metalloproteinase-2, TIMP-2) and renal injury markers (urinary neutrophil gelatinase-associated lipocalin -NGAL)- the bio-sonographic index (BSI).

### Methods

A prospective observational study at a tertiary referral center will be performed enrolling 80 patients undergoing elective open and endovascular repair of the visceral aorta. The BSI will be evaluated at baseline, and at 6 and 24 hours after the procedure. The primary outcome is the occurrence of AKI according to the Kidney Disease Improving Global Outcomes (KDIGO) criteria. Each patient will be his/her own control. A reference group of 15 healthy volunteers who are not undergoing interventions will be enrolled to test the feasibility of and to refine the novel SWE protocol. The BSI will be tested for its predictability of the occurrence of AKI. Comparisons will be made between individual and combined components of the BSI and traditional markers used in the KDIGO definition; serum creatinine and urine

**Data Availability Statement:** All relevant data from this study will be made available upon study completion.

**Funding:** This work was supported by the University of Alabama At Birmingham Surgery Engineering Collaborative Grant Funds 2019. 'Internal funding'. 3122053.000.213122053.311850000.0000 to AZ/AB The funders had and will not play any role in the study design, data collection, and analysis, decision to publish, or the preparation of the manuscript.

**Competing interests:** The authors have declared that no competing interests exist.

output in terms of baseline status of the kidney. Correlations will be made between the BSI and conventional indices of AKI and exploratory analyses will be conducted to identify individual disease patterns using the BSI.

## Discussion

We hypothesize that the BSI will be a sensitive index of early structural abnormalities that precede and predict the occurrence of AKI as defined by KDIGO in complex vascular surgery.

## Trial registration

ClinicalTrials.gov NCT04144894. Registered 1/6/2020.

## Introduction

Postoperative acute kidney injury (pAKI) is a common complication of aortic aneurysm repair occurring in approximately 40% of patients [1–4]. Even in milder forms, pAKI has been associated with progression to chronic kidney disease (CKD), high mortality, morbidity and health care expense. Although patients may recover renal function after acute injury, there is no curative treatment available [5].

A major impediment to the provision of early and effective treatment for pAKI is the inherent delay in diagnosis due to limitations of available markers of kidney function, which are neither sensitive nor specific to the site, laterality, extent, etiology or reversibility of the injury [6–8]. Additionally, current novel biomarkers of renal injury such as neutrophil gelatinase-associated lipocalin (NGAL) suffer from the limitations of being non-specific to the etiology of renal injury, being heavily affected by preoperative kidney function status [9], lack of current consensus on cut off values [10], and the inability to use as therapeutic targets. As such, the diagnosis of pAKI continues to be delayed with consequential delay in initiation of effective and timely therapy and improved outcomes.

Whereas G1 cell cycle arrest (CCA) biomarkers (urinary insulin growth factor binding protein-7, IGFBP-7 and inhibitor of tissue metalloproteinase-2, TIMP-2) have been recently validated as markers of renal stress that predict the occurrence of pAKI [11, 12], they are affected by preoperative disease comorbidities, do not inform on non-tubular nephron pathology, and have been used in such a way that therapy based on their use has been confined to the postoperative period after the occurrence of the injury [13].

Shear wave elastography (SWE) has evolved as an ultrasound tool capable of measuring and quantifying tissue stiffness in response to stress. Tissue stress is applied by emitted U/S beam that generate shear waves, the velocity of propagation of which is measured as a surrogate of deformation (or elastance or stiffness) using Young's Elastic Modulus (YM). The stiffer the tissue, the faster the propagation velocity [14]. Shear wave elastography has been recently used to assess renal tissue stiffness in patients with mild forms of chronic kidney disease (CKD) [15–17]. We hypothesize that detection of renal cortical stiffness may add to the diagnostic value of currently used renal biomarkers.

In this study, we propose a novel index that combines ultrasonography and biomarkers of renal stress and injury to early detect and predict the occurrence of AKI in patients undergoing complex vascular surgery.

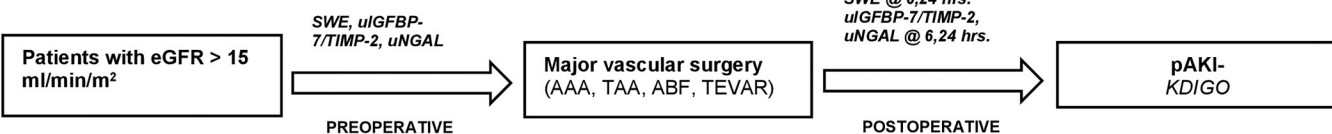

**Fig 1. Proposal summary of the study.** eGR: Estimated glomerular filtration rate, SWE; Shear wave elastography; uIGFBP-2/TIMP-7: urinary Insulin growth factor binding protein-7/tissue inhibitor of metalloproteinase-7; NGAL: neutrophil gelatinase-associated lipocalin; AAA: open Abdominal aortic aneurysm repair; TAA: Thoracoabdominal aneurysm repair; TEVAR: Thoracic endovascular aneurysm repair; PAKI: Postoperative acute kidney injury; KDIGO: Kidney Disease Improving Global Outcomes; EVAR: endovascular abdominal aneurysm repair, FEVAR: fenestrated endovascular abdominal aneurysm repair.

## Methods

### Study design

This is a prospective observational study at a tertiary referral center. The study has been approved by the institutional review board of the University of Alabama at Birmingham. Fig 1 outlines the flow of the study.

### Sample size calculation

Based on our prior experience and literature search of a 40% incidence of pAKI in this population encompassing both open and endovascular approaches of descending aortic aneurysm repairs [1, 18, 19], we expect 40% patients to develop pAKI and 60% of patients to not develop pAKI. We project that more than 70% of patients with a positive BSI to develop pAKI, while only 30% of those with a negative BSI to actually develop pAKI. From this assumption, if we recruit 80 patients to the study, we will have 32 patients expected to have pAKI (positive BSI) and 48 patients expected not to have pAKI (negative BSI). Using Fisher's Exact Test with a 5% two-sided significance level, we will have minimum 90% power to detect the difference between 69% pAKI in patient group who will eventually develop pAKI (positive BSI) and 30% pAKI in patients who will not eventually develop pAKI (negative BSI), when sample sizes are 32 and 48, respectively. In addition, we will recruit 15 healthy group of participants to explore the feasibility of the SWE protocol, and to establish reference values on a sample of our population in terms of renal cortical stiffness, and renal parenchymal and vascular indices.

### Participants

A total of 80 patients for the 'study group' will be enrolled. Each patient will serve as his/her own control by studying the change in BSI from preoperative values to 6 and 24 hr after the procedure. A total of 15 normal healthy volunteers will be enrolled as a 'feasibility group' to test the feasibility of the SWE protocol.

### Study patients

**Inclusion criteria.** Patients undergoing complex vascular surgery defined as open thoracoabdominal aortic aneurysm repairs (TAAA) and thoracic endovascular aneurysm repair (TEVAR), open abdominal aortic aneurysm repairs (AAA) and endovascular abdominal aneurysm repairs (EVAR), fenestrated endovascular abdominal aortic aneurysm repairs (FEVAR) and branched endovascular aortic aneurysm repairs (BEVAR).

**Exclusion criteria.** Patients with estimated glomerular filtration rate $< 15$ ml/min/1.73 $m^2$, any organ transplant, age $< 19$ years, pregnant women, ejection fraction $< 40\%$, body mass index (BMI) $> 30$ kg/m$^2$, emergency procedure, and any condition that would impede

visualization of the kidneys by U/S will be excluded. Also, participants refusing to continue participating in the trial at any time during the conduction of the trial will be excluded.

## Feasibility group

A group of 15 healthy participants with no known history of diabetes mellitus (DM), hypertension or kidney, liver or heart disease and with a BMI < 30 kg/m$^2$ will be enrolled. This group will serve to test the feasibility of SWE protocol and to establish reference values of normality of renal stiffness.

## Informed consent

Consent will be obtained electronically (Red Cap) via an IRB-approved consent forms by any of the research team members.

## Procedures

The bio-sonographic index (BSI) will be performed on the study patients at baseline- during their preoperative visit, and at 6 hours and at 24 hours after the procedure. The BSI will be composed of SWE, urinary cell cycle arrest (uCCA) and urinary NGAL (uNGAL). Abnormal BSI will be diagnosed as an increase of at least 10% from baseline to any of the 6 and 24 hour time points. Based on an estimated incidence of pAKI of 40% in this population, a probability of abnormal BSI will be estimated as follows (Table 1).

**Shear wave elastography exam.** The ultrasonographic examination of the kidney will be performed using the Mindray Rosana 7 system (Mindray North America, CA, USA) equipped by the SC6-1u transducer for acquiring grayscale, spectral Doppler, and virtual touch tissue quantification on acoustic radiation force impulse (ARFI) imaging in the healthy adults. The exam will be performed by a certified radiology technician with over 10 years of experience and read by an independent radiologist who is blinded to the study protocol. We will use the SWE published and outlined by Sandhu et al. [20] as follows: During each encounter, both kidneys will be evaluated. The flank approach will be used to access both kidneys to minimize dissipation of U/S waves by distances traveled through or by pathology of the spleen and liver for the left and right kidneys, respectively. Exams will be performed in the left and right lateral decubitus positions for imaging the left and right kidney, respectively. Before scanning, we will adjust the image acquisition settings as follows: MI 1.4, image depth 10–12 cm, scanning frequency 3.5 MHz, single focus, dynamic range 65, harmonic imaging, Map E/Space-time 2, and total gain 0–1. The presence or absence of hydronephrosis, calculi, masses, and perinephric collections will be also assessed; the presence of any of these findings and will serve as exclusion criteria. For each participant, the following values will be obtained during suspended respiration: each kidney size in the longitudinal plane; main renal artery peak systolic velocity

**Table 1. The BSI criteria.**

| AKI | combination | SWE (YM) | uCCA | uNGAL | Patients | AKI % |
|-----|-------------|----------|------|-------|----------|-------|
| Y | Abnormal | ≥10% = 1, < 10% = 0 | ≥10% = 1 < 10% = 0 | ≥10% = 1 < 10% = 0 | 40% | 90% |
| N | Normal | SWE <10% = 0 | < 10% = 0 | < 10% = 0 | 60% | 30% |

* Values are in reference to a change from baseline to any of time points 2, and 3.

AKI (Y/N) = YM based on SWE & urine biomarkers

AKI: acute kidney injury; SWE: shear wave elastography; YM; Yong Elastic Module; uCCA: urinary cell cycle arrest; uNGAL, urinary neutrophil gelatinase-associated lipocalin; Y: yes; N: no; BSI: biosonographic index; BSI score is the sum of SWE, Biomarker, and NGAL. It is abnormal if ≥ 1. Total score can range from 0 to 3.

(PSV), end diastolic velocity (EDV), and resistive index (RI); inferior interlobular artery PSV, EDV, and RI; mid-interlobular artery PSV, EDV, and RI; superior interlobular artery PSV, EDV, and RI; shear-wave velocity values at the renal cortex (five measurements in the longitudinal plane at the upper, upper-mid, mid, mid-lower, and lower poles and three measurements in the transverse plane at upper, mid, and lower poles). All measurements will be repeated to yield two measurements per parameter per observer. Each patient will be scanned by a technician and by the principle investigator (AZ) on the study. Images will be interpreted independently by a radiologist and a vascular surgeon who are blind to the study protocol.

**Urinary biomarkers NGAL and cell cycle arrest biomarkers.** Urine will be collected at the 3 time points and analyzed for NGAL and CCA biomarkers as the product (IGFBP-7x TIMP-2) (Nephrocheck[R] Test) as previously described in published trials [12, 21] and in accordance with the package insert guidelines. The urine values will be adjusted for albumin and for creatinine based on package insert.

## Perioperative procedures

Patients undergoing interventions on the visceral aorta will be exposed to the standard of care techniques in terms of surgery and anesthesia. For the open procedures (open juxta-, suprarenal AAA and thoracoabdominal [TAAA] repair), patients will undergo general endotracheal anesthesia (GETA) with invasive monitoring in the form of arterial line catheter placement, and central venous line (in AAA procedures) (Table 2). Endovascular techniques (TEVAR, EVAR/FEVAR/BEVAR) will be managed under general endotracheal anesthesia (GETA). Standard and invasive monitors (arterial catheter) will be placed in all, with central venous pressure and PAC catheters placed selectively based on cardiopulmonary morbidities. Given

**Table 2. Anesthetic technique of major vascular procedures at UAB.**

| Anesthetic management | Open TAA | Open AAA | TEVAR | EVAR |
|---|---|---|---|---|
| Anesthesia | GETA with DLT | GETA | GETA | GETA |
| Monitoring | ASA Standard | ASA | ASA | ASA |
| | Rt radial A line | A line | Rt radial A line | A line |
| | Femoral A line | CVC | CVC | |
| | CVC | | Spinal drain | |
| | TEE | | SC monitoring: SSEP/MEP | |
| | Spinal drain | | | |
| | SC monitoring: SSEP/MEP | | | |
| Special considerations | LHB | Clamping/unclamping | Contrast agent | Contrast agent |
| | Clamping/unclamping | | | |
| Hemodynamic goals | **Clamping** | **Clamping** | Goal MAP> 90 mmHg following deployment of endograft | Goal MAP> 90 mmHg following deployment of endograft |
| | Vasodilators: NTG,Short acting beta blockers | Vasodilators: NTG, short acting beta blockers | | |
| | **Unclamping** | **Unclamping** | | |
| | Vasopressors: Norepi | Vaspressors: Norepi | | |
| Anticoagulation | Heparin | Heparin | Heparin | Heparin |
| Disposition | ICU | ICU | ICU | Step down unit |

TAA: thoracoabdominal aortic aneurysm repair; AAA: abdominal aortic aneurysm, TEVAR: thoracic endovascular aneurysm repair; GETA: general endotracheal anesthesia, DLT: double lumen tube; ASA: American Society of Anesthesiologists; Rt: right; CVC: central venous catheter; LHB: left heary bypass; MAP: mean arterial pressure; NTG: nitroglycerine; Norepi: norepinephrine; ICU: intensive care unit; SEEP: somatosensory evoked potential; MEP: motor evoked potential; TEE: transesophageal echocardiography; A-line: arterial line catheter; SC: spinal cord

**Table 3. Renal protection protocol.**

| |
|---|
| **Preoperative** |
| Discontinuation of ACEI/ARB 24 hours preoperatively unless reduced EF or heart failure |
| Preoperative hydration for endovascular procedures (NS 5 ml/kg/min) starting 12 hours preoperatively |
| **Intraoperative** |
| Use of high dilution of contrast agent |
| Intrarenal cold crystalloid (4˚C), mannitol, methylprednisolone |
| IV mannitol 300 mg/kg post unclamping |
| Maintenance of MAP > 65 mmHg |
| **Postoperative** |
| Avoid nephrotoxin |
| Minimize contrast agents |
| Daily monitoring of serum creatinine/UOP |
| Maintenance of blood glucose 140–180 mg/dl |
| Maintenance of MAP > 65mmHg |

ACEI: angiotensin converting enzyme inhibitors, ARB: angiotensin receptor blockers; EF: ejection fraction; IV: intravenous; MAP: mean arterial blood pressure; NS: normal saline; UOP: urine output

the observational nature of the study, conventional anesthetic care, fluid, pharmacologic and blood resuscitation practices will be left to the discretion of the staff anesthesiologist and surgeon. Additionally, organ protection protocols such as spinal cord and kidney protocols will all follow the standard of care at UAB (Tables 3 and 4).

Patients undergoing open aortic repair and F/BEVAR are typically managed in the surgical intensive care unit post-operatively.

## Surgical technique

**Open procedures.** The technique of open repair will vary somewhat with the level of disease and the surgeon performing the operation, and cannot be prescribed by this protocol given the variability in disease. However, all aortic clamp sites, renal ischemia time and methods of renal protection will be documented.

Renal protective maneuvers routinely performed by some surgeons depending on the level of disease and whether a suprarenal clamp will be placed includes ischemic preconditioning with iliac clamping prior to aortic clamping, renal flushing with cold lactated ringers containing solumedrol/mannitol, and systemic mannitol prior to aortic clamp placement [22].

**Endovascular procedures.** The endovascular techniques utilized at UAB (author AWB) depend on the clinical scenario (i.e. urgent vs. emergent repair), the devices utilized (e.g. physician modified vs. custom vs. off the shelf), the aortic pathology being treated (e.g. dissection vs. degenerative aneurysm vs. pseudoaneurysm after prior open/endovascular repair), and the level of disease (i.e. juxtarenal vs. suprarenal vs. thoracoabdominal) and the details of these various scenarios are extensive and outside of the scope of this document. These techniques have been reported in detail previously [23–26].

## Data collected

Demographic information, vital signs, laboratory values, administered medications, intravenous fluids, fluid balance, dose and type of renal contrast, cross clamp duration, mechanical

**Table 4. Spinal cord protection protocol.**

**Preoperative**

• Hold preop blood pressure medication two days prior to procedure for permissive hypertension with exception of beta blockers and clonidine, which should be continued. In patients with heart failure and/or reduced EF, discuss holding ACEI or ARB.

• All patients should be on a statin unless contraindicated

• Preoperative placement of spinal fluid drain: pop-up pressure 10 mmHg, CSF drainage < 20 ml/ hr

**Intraoperative**

• MAP > 65 mmHg with goal of MAP > 90 mmHg following deployment of endograft

• Optimization of cardiac index with vasopressors and/or inotropes

• IV naloxone drip (1 ug/kg/hr) started at beginning of case and continued for 48 hours

• Avoid long-acting narcotics (Morphine and Hydromorphone)

• Insulin drip to maintain glucose < 200 mg/dL

• Mannitol 12.5 gm prior to graft deployment and 12.5 gm after graft deployment if issues with patency of the spinal drain. Mannitol avoided if CSF drain is working well.

• Goal hemoglobin of 10 g/dL

• Mild hypothermia

**Postoperative**

• Continue insulin drip to maintain glucose < 200

• Passive rewarming in patients with mild hypothermia (>34C)

• Continue naloxone drip for 48 hours

• MAP> 90 mmHg and heart rate < 90 for 48 hours or until spinal drain removed

• Avoid arterial dilators (nitroprusside, hydralazine, milrinone) for treatment of hypertension

• CVP >10 mmHg for ventilated patients. CVP >7 mmHg for nonventilated patients

• Goal hemoglobin > 9 g/dL for first 5 days post op. Goal hemoglobin ≥10 g/dL if evidence of SCI. After 5 days if the patient is without SCI symptoms, hemoglobin goal decreased to >7 g/dL.

• CSF drainage for 24 hours at popoff of 10 mmHg. Drain clamped at 24 hours if no SCI symptoms and remains clamped for 18–24 hours with q1 hr neuro checks.

• D/C spinal drain after 72 hours if no evidence of SCI.

• Spinal drain left in place for at least 72 hours after the onset of SCI with a popoff of 10 mmHg. Spinal pressure may be lowered to alleviate symptoms.

**Rescue Maneuvers if SCI**

• If spinal drain not in place, emergent drain placement requested

• If spinal drain in place, decrease the popoff to 5 mmHg. Do not drain more than 40 mL/hr of CSF

• Transfuse to goal hemoglobin ≥ 10 g/dL

• Mannitol 12.5 grams IV over 15 minutes

• Increase goal MAP >100 mmHg unless contraindicated

• Goal cardiac index > 2.5

• Methylprednisolone 1000mg IV infusion over 30 minutes

• If not already infusing begin naloxone infusion at 1–1.5 ug/Kg/hr

MAP: mean arterial blood pressure; SC: spinal cord; SCI: spinal cord injury; CVP: central venous pressure; CSF: cerebrospinal fluid

ventilation data and durations will be collected at baseline and throughout the patients' hospital stay in the study group.

The intraoperative anesthetic and postoperative intensive care management of these patients will be the responsibility of the anesthesiologist or critical care physician, respectively, and will be according to the standard of care provided for these patients.

## Outcome

The outcome is the development of AKI based on changes in serum creatinine and urine output according to Kidney Disease Improving Global Outcomes (KDIGO) criteria [27].

## Statistical analyses

The analysis plan will include comprehensive graphical and statistical descriptions of the data. Baseline demographics, biomarkers, stratified by pAKI status will be summarized using descriptive summary statistics. Continuous variables will be summarized overall and within each group with the total numbers of observations, means with 95% confidence intervals, standard deviations and ranges. The two-sample t-test will be employed to compare the mean of continuous outcome variables while the Wilcoxon rank-sum test will be used for study outcomes that do not meet the normality assumption after performing appropriate data transformations. Categorical variables will be summarized overall and within each group with the total numbers of observations and percentages with exact 95% confidence intervals. To assess the accuracy and concordance of each marker with the pAKI status, we will calculate sensitivity and specificity. We will evaluate concordance (sensitivity and specificity) between pAKI status and SWE, uCCA, uNGAL alone and in combination. The BSI accuracy in predicting pAKI status will be evaluated in this study. To investigate the prognostic utility of bio-sonographic index for prediction of pAKI status, we will compare the sensitivity and specificity, using the 10% change from baseline to any of the 6 and 24 hour postoperative values alone as well as combined score (range: 0–3). Receiver operating curve (ROC) will be constructed for comparison between individual and combined components of SWE, uCCA, and uNGAL, and the differences in areas under the curve (AUC) will be assessed using a critical "z" ratio proposed by Hanley and McNeil. Intra-subject variability will be examined by comparing difference of individual readings from the median YM for each subject. The diagnostic performance of SWE and combination of each for distinguishing normal renal parenchyma from renal parenchyma affected by AKI will be assessed using a univariate logistic regression model to construct receiver operating characteristic (ROC) curves. ROC analyses will be performed to determine a cutoff point of SWE that would correctly classify the maximum number of participants based on sensitivity and specificity values. Wald asymptotic 95% confidence limits will be presented for sensitivity and specificity values. Percentile method 95% confidence intervals for ROC curves will be generated using the pROC package in R version 3.0.2 (Vienna, Austria) [28, 29]. Comparisons of area under the curves will be performed by use of a contrast matrix to take differences of the area under the empirical ROC curves. Pearson correlation coefficients will be used to assess the strength of association between continuous exposure variables. Stratified analysis of the control and AKI groups will be performed to evaluate for potential significant confounders of SWE values. SAS version 9.4 (Cary, NC) will be used for all other non-ROC related statistical analysis. Two-tailed p values of less than 0.05 will be deemed statistically significant.

## Intra- and interobserver reliability

Intra- and inter-observer values/data will be evaluated for repeatability (between subjects) and reproducibility (between readers). Average values of will be quantified and Bland–Altman analysis will be performed to assess the repeatability of the techniques, while the Friedman test will be used to assess reproducibility.

### Exploratory analyses

**Vascular resistance-cortical stiffness relationship in pAKI.** The relationship between renal vascular indices in the form of renal resistive index as measured by Doppler imaging and renal cortical stiffness as assessed by SWE will be explored at each of the 3 time points using the Kruskal–Wallis test.

**Cortical stiffness-biomarker relationship in pAKI.** The relationship between the SWE on one side and urinary CCAs and NGAL (biomarkers) values on the other side will be explored to detect the relationship between renal stiffness and markers of tubular stress and injury at the 3 above-mentioned time points.

**Cortical stiffness-biomarker-biochemical relationship in pAKI.** The relationship between SWE, biomarkers and serum creatinine across the above-mentioned 3 time points will be explored to compare which of the parameters is capable of detecting the earliest change in renal pathology.

### Disease patterns in AKI using BSI

An exploratory analysis will be performed to attempt to characterize whether there are specific patterns for diabetes, hypertension or both before and after surgery.

### Data handling and security

Data safety oversight will be provided by the Data and Safety Monitoring Board (DSMB). The DSMB board committee will be independently conducting internal audits to ensure the proper conduct of the trial.

### Mechanisms for HIPAA compliance

Personal identifiers will be protected by the investigators during the course of data collection. All personnel will adhere to HIPPA guidelines and have successfully completed UAB IRB training in the protection of human subjects involved in research. Data will be entered into a database on a password protected computer of the Department of Anesthesiology's information system. All data containing identifiers are already maintained in an IRB approved database. Therefore, the data will not be destroyed at the end of the study. If any data is obtained directly from the medical records as part of this study, it will be entered into an electronic database. This database will be destroyed once the study is complete.

Data safety oversight will be provided by the Data and Safety Monitoring Board (DSMB). The DSMB board committee will be independently conducting internal audits to ensure the proper conduct of the trial.

### Dissemination policy

The results of this study will be communicated to the public through publications with no restrictions. No public access will be allowed to the data of this study.

### Adverse events reporting

This study is observational in nature with no interventions. All adverse events will be reported to the DSMB.

## Discussion

To our knowledge this would be the first study that combines biomarkers and ultrasound for predicting pAKI in the vascular surgical population.

The novelty of this study does not only stem from developing a novel index for pAKI detection and prediction, but in highlighting some of the limitations of the current AKI definition that contribute to overlooking a timely diagnosis of AKI and institution of interventions in the vascular surgical population.

Patients undergoing endovascular aortic aneurysm repairs are characterized by a delayed presentation of kidney decline that may contribute to an overall worse outcome [30]. As such, it is unknown whether this is an ongoing process of kidney damage or a new yet delayed onset kidney decline. By introducing a sensitive index of kidney structural decline, the BSI has the potential of identifying kidney decrement at an early and potentially reversible phase.

Combining an imaging tool with biomarkers of tubular stress and damage has the potential to increase the specificity of novel renal biomarkers by providing a 'renal' origin of the elevated biomarker.

The BSI has the potential of identifying baseline renal damage currently overlooked by normal serum creatinine and estimated glomerular filtration rate. This may better classify vascular surgical patients in terms of baseline kidney status. Our plans to adjust measured biomarkers for urine protein overcomes some of the limitations of previous studies that did not perform such an adjustment [31].

We chose a percent change in the components of the BSI rather than predetermined cut off values for several reasons. First, there is no consensus on SWE values in pAKI and hence we believe that a 10% increase in renal cortical stiffness may be a reasonable of a change from baseline. Second, there is no consensus on baseline preoperative cut off values for the majority of the widely used renal biomarkers. Therefore, assessing a change rather than cut off values may be a reasonable strategy. Third, except for uCCA biomarkers, there is no consensus on a cut off value for the prediction of pAKI. Fourth, with respect to uCCA, there is recent evidence to suggest that values below 0.3 (>0.8, <0.3) could be predictive of AKI [32]. Collectively, given the exploratory nature of the study, and in order to achieve consistency between the components of the BSI, a change of 10% from baseline was chosen an indicator of pathology. The prediction of such a value remains to be determined with further validation of the index.

We are aware of the potential limitations of the SWE for the assessment of the kidneys. SWE was mainly developed to target isotropic planar organs that are superficial and more accessible for U/S probe. The native kidney is highly anisotropic and deeply seated. In order to overcome the caveat of deep seating, the flank approach will be used to minimize interference from liver and spleen with path of U/S elastography signal. Furthermore, patients with low BMI and no prohibitive technical impedances to U/S will be enrolled. While renal anisotropy may be regarded as a limitation to SWE, it could be a potential advantage; the sensitivity of SWE to different renal tissue elasticities may be able to detect sites with early and more advanced pathologies within the same kidney, a phenomenon that is analogous to different types of strain in the heart [33]. Shear wave elastography may be sensitive to increased vascularity of the kidney, a common occurrence after vascular surgery. We plan to overcome refine the technique to differentiate the change from pre-procedural exam. Whereas tissue fibrosis has been the main indication to use SWE, our exploratory study will extend the use of SWE to detect tissue stiffness that is not due to fibrosis in an acute setting of renal pathology. We hypothesize that there will be acute changes in renal vasculature and parenchyma that will lead to an increase in cortical stiffness in an acute setting such as increased vascularity from inflammation, or increased back pressure from the heart as a result of worsening perioperative

cardiac diastolic function [34]. Furthermore, our renal U/S examination images other aspects of the kidneys in terms of RRI, size and size, all may be proven helpful to observe in an acute setting. Additionally, SWE has been used to assess tissue stiffness that is not due to fibrosis in other organs such as breasts [14, 35].

Data obtained from this study will be used to evaluate the performance of other biomarkers within this index and will be validated in an adequately powered randomized clinical trial.

## Author Contributions

**Conceptualization:** Ahmed Zaky, Adam W. Beck, Adam Sturdivant, Novak Zdenek, Brad Meers, Michelle Robbin, J. F. Pittet, Ashita Tolwani, Dan Berkowitz.

**Data curation:** Ahmed Zaky, Sejong Bae, Adam Sturdivant, Nicole McAnally, Ashita Tolwani.

**Formal analysis:** Ahmed Zaky, Sejong Bae, Nicole McAnally, Ashita Tolwani.

**Funding acquisition:** Ahmed Zaky, Adam Sturdivant, Ashita Tolwani, Dan Berkowitz.

**Investigation:** Ahmed Zaky, J. F. Pittet, Ashita Tolwani, Dan Berkowitz.

**Methodology:** Ahmed Zaky, Adam W. Beck, Sejong Bae, Adam Sturdivant, Amandiy Liwo, Novak Zdenek, Shama Ahmad, J. F. Pittet, Ashita Tolwani, Dan Berkowitz.

**Project administration:** Ahmed Zaky, J. F. Pittet.

**Resources:** Adam W. Beck, Sejong Bae, Nicole McAnally, Michelle Robbin, J. F. Pittet, Ashita Tolwani, Dan Berkowitz.

**Software:** Sejong Bae.

**Supervision:** Ahmed Zaky, Michelle Robbin, J. F. Pittet, Ashita Tolwani, Dan Berkowitz.

**Writing – original draft:** Ahmed Zaky, Adam W. Beck, Sejong Bae, Adam Sturdivant, Amandiy Liwo, Shama Ahmad, Brad Meers, J. F. Pittet, Ashita Tolwani, Dan Berkowitz.

**Writing – review & editing:** Ahmed Zaky, Adam W. Beck, Adam Sturdivant, Amandiy Liwo, Novak Zdenek, Nicole McAnally, Shama Ahmad, Brad Meers, J. F. Pittet, Ashita Tolwani, Dan Berkowitz.

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
