## [Decision Letter · Decision Letter 0]

23 Jul 2020

PONE-D-20-08584

The Bio-sonographic Index. A Novel Modality for Early Detection of Acute Kidney Injury after Complex Vascular Surgery. A Protocol for an Exploratory Prospective Study

PLOS ONE

Dear Dr. Zaky,

Thank you for submitting your manuscript to PLOS ONE. After careful consideration, we feel that it has merit but does not fully meet PLOS ONE’s publication criteria as it currently stands. Therefore, we invite you to submit a revised version of the manuscript that addresses the points raised during the review process.

Specifically, the reviewers had overlapping concerns about the study design and proposed statistical methodology presented in the manuscript.

We look forward to receiving your revised manuscript.

Kind regards,

Richard Hodge

Associate Editor

PLOS ONE

Journal Requirements:

Additional Editor Comments (if provided):

Reviewers' comments:

Reviewer's Responses to Questions

**Comments to the Author**

1. Does the manuscript provide a valid rationale for the proposed study, with clearly identified and justified research questions?

Reviewer #1: Partly

Reviewer #2: Yes

Reviewer #3: Yes

2. Is the protocol technically sound and planned in a manner that will lead to a meaningful outcome and allow testing the stated hypotheses?

Reviewer #1: Partly

Reviewer #2: Yes

Reviewer #3: Yes

3. Is the methodology feasible and described in sufficient detail to allow the work to be replicable?

Reviewer #1: Yes

Reviewer #2: Yes

Reviewer #3: Yes

4. Have the authors described where all data underlying the findings will be made available when the study is complete?

Reviewer #1: Yes

Reviewer #2: No

Reviewer #3: Yes

5. Is the manuscript presented in an intelligible fashion and written in standard English?

Reviewer #1: Yes

Reviewer #2: Yes

Reviewer #3: Yes

6. Review Comments to the Author

You may also provide optional suggestions and comments to authors that they might find helpful in planning their study.

Reviewer #1: The authors attempt to establish a modality for early detection of acute kidney injury after complex vascular surgery and exhibit the protocol in this manuscript. I have following major comments:

1. Based on the observational nature of the study, the authors described in the section of perioperative procedures ”conventional anesthetic care, fluid, pharmacologic and blood resuscitation practices will be left to the discretion of the staff anesthesiologist and surgeon”. Multiple factors might be associated with pAKI, therefore the authors would provide the details of the perioperative procedures so that to confirm the methods difference would not affect the primary endpoint. Furthermore, the difference of the perioperative procedures might influence the sample size estimation. In a word, the authors would provide the protocol of the perioperative procedures adopted in the manuscript.

2. Authors used traditional marker “creatinine” as the “positive control” to help evaluating the new biomarker “IGFBP-7/TIMP-2”. Given the nature that “creatinine” was the marker of renal function, adding new biomarker related with renal injury, such as NGAL/Kim1 would be better.

Reviewer #2: Dr. Zaky and colleagues outline a proposal to evaluate a novel modality for early detection of AKI. I think this is a great idea that addresses a clear need, surgery-associated AKI prediction and early detection. However, I found some of the details a bit confusing. My intent here is to ask clarifying questions to improve the manuscript, but I like the overall concept of the study.

1. At various points in the manuscript, the goal seems to vacillate from prediction and early detection of AKI to redefinition of AKI entirely. While that is a worthy long-term goal given the limitations of the current KDIGO-based definition, it is outside the scope of this project. The abstract says “an approach that is based on structural characterization of AKI is more specific and sensitive to its occurrence” for instance. I mention it here in the abstract, but it occurs at other points, and I would make sure the focus of prediction and early detection is consistent throughout.

2. I don’t think the rationale behind obtaining three different shear wave elastography measurements within hours of each other is clear. I looked at the references provided (10-13), and they all measure shear wave elastography in the setting of static processes such as fibrosis and tumor density, and not for dynamic processes such as inflammation, edema, vascular changes, necrosis, apoptosis, or other acute sorts of changes. Why do you expect to see changes in the your elastography findings, given that fibrosis and other major structural remodeling don’t happen within the timeframe proposed? It seems to me that the biggest benefit of using elastography prior to surgery is that you may identify patients with underlying fibrosis, suggestive of early “CKD” that has not yet reached an eGFR < 60 (as noted in the Introduction). I don’t see the value of the post-surgical SWE, however.

a. Minor point: I saw that the authors in their limitations section note they will assess stiffness not due to fibrosis, and cited (10). That was a bit misleading, because that study is still looking at tumor stiffness compared to normal tissue at a single point in time, rather than any sort of dynamic process.

b. Minor point: in the exploratory analyses, it notes exploring relationship between CCAs and fibrosis, which doesn’t make much physiologic sense to me. As far as I know, these CCAs don’t have much utility in CKD detection, and I’m not sure why they would based on the mechanism. The section continues talking about renal cotical tissue fibrosis changes at 3 different time points. There’s no pathophysiological basis that I can see there, either, since kidney fibrosis does not change significantly over a 6 hour period, or in the immediate post-op period as proposed. Even though these are exploratory aims, the rationale behind them needs to be clear.

c. Minor point: You say in the limitations section that you will avoid CKD, but you only exclude GFR < 15 in the exclusions section, so are CKD 3 and 4 eligible, or not?

d. As for post-surgical studies, have you considered other experimental US techniques, such as those measuring cortex microvascular flow? You have a number of exploratory aims, so just a thought.

3. The purpose of the control group is unclear to me. Standard values (4.7 kPa) exist already, and are the basis for your BSI. There is a line about evaluating for potential confounders of SWE values, although I don’t know that the controls will actually be useful in that regard. Also, are the controls getting 3 sets of US and CCA markers drawn, like in the surgery group?

4. Table 1 says 90% for AKI in patients with positive tests, while the power calculations use 70%.

5. How the two parts of the BSI work together as part of a single score is unclear to me. You have a cutoff for SWE and a cut-off for CCAs. Is BSI binary, such that having one positive (SWE or CCA) make the whole BSI positive? Is there a score that is generated?

6. For point 3 in the discussion section, I’m not sure how you are overcoming the limitations of creatinine, since you are still using creatinine to define AKI, rather than advancing a new definition of AKI (see point 1 above).

a. Sub-point: I don’t see how this metric will overcome under-representation of women and elderly in longitudinal studies, which the manuscript seems to be imply the BSI will do. I would remove the part about direct measurement of GFR, which doesn’t seem to have much to do with this study.

7. I don’t understand the fourth discussion comment about proteinuria. Are you referring to how you are adjusting the CCA urine values for albumin and creatinine? That may not have been done in prior studies, but it doesn’t seem to be a unique feature of the BSI. You could adjust for albumin without using SWE.

Reviewer #3: The study aims to use novel index (ultrasonography and biomarkers) to early detect and predict the occurrence of AKI in patients undergoing complex vascular surgery.

It is a good study to explore and if successful will have implication on the diagnosis method.

However, the manuscript requires further improvement.

Comments

Sample size calculation

Page 4 Paragraph 1, what abnormal patient and normal patient group refers to be clearly stated.

Page 5 Table 1, symbol <= to be replaced with symbol ≤

Procedures

Page 5, Kpa to be standardized kPa.

It would be good to have a chart to indicate/summarize the type of assessment/measurement according to the assessment period including number of times of measurement.

Statistical analyses

Page 9, for the statement 'Pearson correlation coefficients will be used to assess the ‘association’ between continuous exposure variables’ to be revised as to assess the ‘strength of association’.

Page 9 Paragraph 1, typo 4.7kP

Page 10, the description on the exploratory analyses of 'Vascular-parenchymal relationship in pAKI', 'Radiological-biomarker relationship in pAKI', 'Radiological-biomarker-biochemical relationship in pAKI' and statistical analyses' not clear and requires revision. What 3 times points refers to be clearly stated (i.e. pre and post or at each time point)

Page 10, the write-up can be further improved by avoiding the use of ‘we plan’ too many times and could be written in passive form.

Data handling and security

Page 11, the sentence ‘Oversight of this investigation will be provided’ to be revised.

Figure 1 requires improvement. The figure is difficult to be visualized.

References to conform with the journal format.

7. PLOS authors have the option to publish the peer review history of their article (what does this mean?). If published, this will include your full peer review and any attached files.

Reviewer #1: No

Reviewer #2: No

Reviewer #3: No

---

## [Author Response · Author response to Decision Letter 0]

14 Aug 2020

Comments to the Author

1. Does the manuscript provide a valid rationale for the proposed study, with clearly identified and justified research questions?

Reviewer #1: Partly

Reviewer #2: Yes

Reviewer #3: Yes

2. Is the protocol technically sound and planned in a manner that will lead to a meaningful outcome and allow testing the stated hypotheses?

Reviewer #1: Partly

Reviewer #2: Yes

Reviewer #3: Yes

3. Is the methodology feasible and described in sufficient detail to allow the work to be replicable?

Reviewer #1: Yes

Reviewer #2: Yes

Reviewer #3: Yes

4. Have the authors described where all data underlying the findings will be made available when the study is complete?

Reviewer #1: Yes

Reviewer #2: No

Reviewer #3: Yes

5. Is the manuscript presented in an intelligible fashion and written in standard English?

Reviewer #1: Yes

Reviewer #2: Yes

Reviewer #3: Yes

6. Review Comments to the Author

You may also provide optional suggestions and comments to authors that they might find helpful in planning their study.

Reviewer #1: The authors attempt to establish a modality for early detection of acute kidney injury after complex vascular surgery and exhibit the protocol in this manuscript. I have following major comments:

1. Based on the observational nature of the study, the authors described in the section of perioperative procedures ”conventional anesthetic care, fluid, pharmacologic and blood resuscitation practices will be left to the discretion of the staff anesthesiologist and surgeon”. Multiple factors might be associated with pAKI, therefore the authors would provide the details of the perioperative procedures so that to confirm the methods difference would not affect the primary endpoint. Furthermore, the difference of the perioperative procedures might influence the sample size estimation. In a word, the authors would provide the protocol of the perioperative procedures adopted in the manuscript.

Authors’ response: we thank the reviewer for his comment. We have addressed our anesthesia and organ protection protocols in Tables 2-4 

2. Authors used traditional marker “creatinine” as the “positive control” to help evaluating the new biomarker “IGFBP-7/TIMP-2”. Given the nature that “creatinine” was the marker of renal function, adding new biomarker related with renal injury, such as NGAL/Kim1 would be better.

Authors’ response

We have added NGAL to the BSI as suggested by the reviewer. 

Reviewer #2: Dr. Zaky and colleagues outline a proposal to evaluate a novel modality for early detection of AKI. I think this is a great idea that addresses a clear need, surgery-associated AKI prediction and early detection. However, I found some of the details a bit confusing. My intent here is to ask clarifying questions to improve the manuscript, but I like the overall concept of the study.

1. At various points in the manuscript, the goal seems to vacillate from prediction and early detection of AKI to redefinition of AKI entirely. While that is a worthy long-term goal given the limitations of the current KDIGO-based definition, it is outside the scope of this project. The abstract says “an approach that is based on structural characterization of AKI is more specific and sensitive to its occurrence” for instance. I mention it here in the abstract, but it occurs at other points, and I would make sure the focus of prediction and early detection is consistent throughout.

Authors’ response

We thank the reviewer for his comment. We have made substantial changes in the wording to consistently reflect the reviewer’s suggestion (Abstract, P:1, L: 28-30, P:3, L:6-7; P:13, L:19-23)

2. I don’t think the rationale behind obtaining three different shear wave elastography measurements within hours of each other is clear. I looked at the references provided (10-13), and they all measure shear wave elastography in the setting of static processes such as fibrosis and tumor density, and not for dynamic processes such as inflammation, edema, vascular changes, necrosis, apoptosis, or other acute sorts of changes. Why do you expect to see changes in the your elastography findings, given that fibrosis and other major structural remodeling don’t happen within the timeframe proposed? It seems to me that the biggest benefit of using elastography prior to surgery is that you may identify patients with underlying fibrosis, suggestive of early “CKD” that has not yet reached an eGFR < 60 (as noted in the Introduction). I don’t see the value of the post-surgical SWE, however.

Authors’ response

We thank the reviewer for his comment. Whereas, SWE has been used to detect fibrosis in a static setting, there are other factors that may lead to cortical stiffness that are not related to fibrosis and that may occur in an acute dynamic setting, such as an increase in vascularity resulting from inflammation, an increase in back pressure from the heart due to perioperative worsening of diastolic function. furthermore, we will be exploring non-stiffness renal U/S that could be of diagnostic value in an acute dynamic setting. By having each patient as his/her control we plan to identify changes in renal stiffness after the procedure compared to before procedure. Since this is the first study to assess renal stiffness in AKI, our current hypothesis is exploratory in nature. We have substantiated this concept in the discussion (P: 18, L:9-18), added 2 references (ref., 29 and 30) 

a. Minor point: I saw that the authors in their limitations section note they will assess stiffness not due to fibrosis, and cited (10). That was a bit misleading, because that study is still looking at tumor stiffness compared to normal tissue at a single point in time, rather than any sort of dynamic process.

b. Minor point: in the exploratory analyses, it notes exploring relationship between CCAs and fibrosis, which doesn’t make much physiologic sense to me. As far as I know, these CCAs don’t have much utility in CKD detection, and I’m not sure why they would based on the mechanism. The section continues talking about renal cotical tissue fibrosis changes at 3 different time points. There’s no pathophysiological basis that I can see there, either, since kidney fibrosis does not change significantly over a 6 hour period, or in the immediate post-op period as proposed. Even though these are exploratory aims, the rationale behind them needs to be clear.

Authors’ response

We thank the reviewer for his comments. As indicated, SWE measures tissue stiffness of which fibrosis is only a cause. There are other causes for renal stiffness that are not due to fibrosis such as right heart diastolic dysfunction, increased tissue vascularity or an increase inflammatory response, all are factors that we would expect to observe on the SWE exam as a dynamic change from pre to postoperatively. As well, we will be observing non-stiffness-related indicators of renal inflammation in terms of vascularity and parenchyma that may all provide an additional diagnostic value in an acute setting. Exploring the relationship between renal cortical stiffness and CCA aims at exploring a relationship between radiological and biochemical evidence of inflammation at the renal cortex. As such, this is an exploratory analysis that may show a correlation. We have substantiated this concept in the discussion (P: 18, L:9-18), added 2 references (ref., 29 and 30)

c. Minor point: You say in the limitations section that you will avoid CKD, but you only exclude GFR < 15 in the exclusions section, so are CKD 3 and 4 eligible, or not?

Authors’ response

We thank the reviewer for his comments. we will include patients with eGFR > 15 ml/min/m2 including CKG stages 3 and 4. Only CKD stage 5 will be excluded.

d. As for post-surgical studies, have you considered other experimental US techniques, such as those measuring cortex microvascular flow? You have a number of exploratory aims, so just a thought.

Authors’ response

we thank the reviewer for his comments. We have considered the use of contrast enhanced renal U/S to assess microcirculation, yet the process of getting approval for a human study is long since it is not FDA approved for this indication as of yet. Furthermore, the technique is not feasible clinically as it is time consuming, exposes patients to contrast risks, is expensive and requires a high level of expertise that may interfere with flow dynamics of the study.

3. The purpose of the control group is unclear to me. Standard values (4.7 kPa) exist already, and are the basis for your BSI. There is a line about evaluating for potential confounders of SWE values, although I don’t know that the controls will actually be useful in that regard. Also, are the controls getting 3 sets of US and CCA markers drawn, like in the surgery group?

Authors’ response

Each patient will serve as his own control. We have changed the name of the control group into the feasibility group which will be studied to determine feasibility of performing and refining the SWE protocol and establishing reference values for normality in a sample of our population. We have also changed cutoff from an absolute value to a % change from baseline, (P:10, L:17-20)

4. Table 1 says 90% for AKI in patients with positive tests, while the power calculations use 70%.

Authors’ response

Power calculation is assuming lower % of AKI (70%) in patients we should be able to detect for a given sample size. If we have 90% for AKI, we will have higher power. 

5. How the two parts of the BSI work together as part of a single score is unclear to me. You have a cutoff for SWE and a cut-off for CCAs. Is BSI binary, such that having one positive (SWE or CCA) make the whole BSI positive? Is there a score that is generated?

Authors’ response

We have added NGAL to the BSI and given the lack of consensus on NGAL abnormal values, have unified the BSI as a % change from baseline. We have adjusted statistics accordingly (P:10, l:2-20)

6. For point 3 in the discussion section, I’m not sure how you are overcoming the limitations of creatinine, since you are still using creatinine to define AKI, rather than advancing a new definition of AKI (see point 1 above).

a. Sub-point: I don’t see how this metric will overcome under-representation of women and elderly in longitudinal studies, which the manuscript seems to be imply the BSI will do. I would remove the part about direct measurement of GFR, which doesn’t seem to have much to do with this study.

Authors’ response

We thank the reviewer for his comment. We have removed that section form the discussion as suggested, have adjusted the discussion to align with the flow of the manuscript (Abstract, P:1, L: 28-30, P:3, L:6-7; P:13, L:19-23)

7. I don’t understand the fourth discussion comment about proteinuria. Are you referring to how you are adjusting the CCA urine values for albumin and creatinine? That may not have been done in prior studies, but it doesn’t seem to be a unique feature of the BSI. You could adjust for albumin without using SWE.

Authors’ response

We agree with the reviewer and have better explained the benefits of adjusting for urine albumin when reporting CCA values and interpretations (P:14, L:11-13).

Reviewer #3: The study aims to use novel index (ultrasonography and biomarkers) to early detect and predict the occurrence of AKI in patients undergoing complex vascular surgery.

It is a good study to explore and if successful will have implication on the diagnosis method.

However, the manuscript requires further improvement.

Comments

Sample size calculation

Page 4 Paragraph 1, what abnormal patient and normal patient group refers to be clearly stated.

Authors’ response

We apologize for the confusion. We have explained the statement more clearly (P 4, L5-11)

Page 5 Table 1, symbol <= to be replaced with symbol ≤

Procedures

Page 5, Kpa to be standardized kPa.

It would be good to have a chart to indicate/summarize the type of assessment/measurement according to the assessment period including number of times of measurement.

Authors’ response

Changes made as suggested

Statistical analyses

Page 9, for the statement 'Pearson correlation coefficients will be used to assess the ‘association’ between continuous exposure variables’ to be revised as to assess the ‘strength of association’.

Page 9 Paragraph 1, typo 4.7kP

Authors’ response

Changes made as suggested

Page 10, the description on the exploratory analyses of 'Vascular-parenchymal relationship in pAKI', 'Radiological-biomarker relationship in pAKI', 'Radiological-biomarker-biochemical relationship in pAKI' and statistical analyses' not clear and requires revision. What 3 times points refers to be clearly stated (i.e. pre and post or at each time point)

Authors’ response

We apologize for the lack of clarity. We have rewritten the exploratory analyses referred to and clarified the 3 time points. (P:11, L: 1-15)

Page 10, the write-up can be further improved by avoiding the use of ‘we plan’ too many times and could be written in passive form.

Page 9 Paragraph 1, typo 4.7kP

Authors’ response

Changes made as suggested 

Data handling and security

Page 11, the sentence ‘Oversight of this investigation will be provided’ to be revised.

Figure 1 requires improvement. The figure is difficult to be visualized.

References to conform with the journal format.

Authors’ response

We have changed the statement as suggested (P:12, L:21-23)

We have also modified the figure and made the change in the legend

7. PLOS authors have the option to publish the peer review history of their article (what does this mean?). If published, this will include your full peer review and any attached files.

Do you want your identity to be public for this peer review? For information about this choice, including consent withdrawal, please see our Privacy Policy.

Reviewer #1: No

Reviewer #2: No

Reviewer #3: No

---

## [Decision Letter · Decision Letter 1]

30 Sep 2020

PONE-D-20-08584R1

The Bio-sonographic Index. A Novel Modality for Early Detection of Acute Kidney Injury after Complex Vascular Surgery. A Protocol for an Exploratory Prospective Study

PLOS ONE

Dear Dr. Zaky,

Thank you for submitting your manuscript to PLOS ONE. After careful consideration, we feel that it has merit but does not fully meet PLOS ONE’s publication criteria as it currently stands. Therefore, we invite you to submit a revised version of the manuscript that addresses the points raised during the review process.

We would particularly like you to address the comments made by Reviewer 1, some of which may be addressed by expanding the discussion to include some of the points raised regarding potential limitations of the study. 

We look forward to receiving your revised manuscript.

Kind regards,

Nicholas M Selby, BMedSci BMBS MRCP DM

Academic Editor

PLOS ONE

Reviewers' comments:

Reviewer's Responses to Questions

**Comments to the Author**

1. Does the manuscript provide a valid rationale for the proposed study, with clearly identified and justified research questions?

Reviewer #1: Yes

Reviewer #2: Yes

2. Is the protocol technically sound and planned in a manner that will lead to a meaningful outcome and allow testing the stated hypotheses?

Reviewer #1: Yes

Reviewer #2: Yes

3. Is the methodology feasible and described in sufficient detail to allow the work to be replicable?

Reviewer #1: Yes

Reviewer #2: Yes

4. Have the authors described where all data underlying the findings will be made available when the study is complete?

Reviewer #1: Yes

Reviewer #2: Yes

5. Is the manuscript presented in an intelligible fashion and written in standard English?

Reviewer #1: Yes

Reviewer #2: Yes

6. Review Comments to the Author

You may also provide optional suggestions and comments to authors that they might find helpful in planning their study.

Reviewer #1: the manuscript provide a valid rationale for the proposed study, with clearly identified and justified research questions. I have no additional comments about this manuscript

Reviewer #2: Thank you for allowing me to review the changes made by Dr. Zaky et. al in their protocol. I commend the authors on their revisions, and I think this version is an improvement. I continue to think the overall aims of the project are valuable, and I have only a few comments that I hope will help with the manuscript.

BSI Components

1. I appreciate the further clarification of the BSI components (10% increase from pre-surgical values in either SWE, Nephrocheck, or NGAL). However, there is no data given to support these cut-points, and it appears to me that this study is expected to generate some of that data. If it is indeed the case that the cutpoints might change based on your findings, it might be useful to explain how you intend to approach creation of the BSI score in the methods, rather than giving specific cut points a priori.

2. I know UAB is a leader in NGAL use, but I would be concerned about using that test in this population. NGAL is well known to increase in high-inflammatory states due to production from extra-renal sources. NGAL is likely to go up simply from post-surgical inflammation. I see this was added based on Reviewer #1’s recommendation, but I suspect it will not be a useful addition, personally. I suppose given that this study is designed to optimize the BSI, you could see whether NGAL adds value to the BSI or not.

3. Nephrocheck has a fairly standard value of 0.3 (as in the PREV-AKI trial, for instance). The 10% change is a metric I’ve not seen before with that marker.

4. I continue to be skeptical that there will be significant SWE changes that occur as early as 6 to 24 hours, but I suppose we will find out. I appreciate the inclusion of additional references and discussion.

Sample Size Calculation

5. 40% strikes me as a very high estimate of post-operative AKI, especially given the study design, and doesn’t seem supported by the references provided.

a. Reference 1, Hobson et al.: EVAR: 5.5% – 18%; Branched or fenestrated AAA: 28%; TEVAR: 9.7% (30% after branched aortic dissections); Huber et al. (cited within reference 1) had an AKI rate of 49%, but this seems to be driven by a high number of emergency cases. Emergency cases are an exclusion criteria in this study.

b. Reference 2, Martin-Gonzalez et al. Fenestrated and Branched Endografting: 29% AKI

c. Reference 3, Lee et al; Snorkel approach for EVAR: 32.6% AKI

d. Reference 4, Saratiz et al. Elective EVAR: 18.8% AKI

e. We recently published a study showing a 20.8% AKI rate following elective or urgent aneurysm repair, and several other studies I pulled up in a brief lit search showed AKI rates closer to 15-25% depending on the exact procedure (in elective cases).

My overall concern is that this study will be underpowered with only 80 patients.

6. The language regarding “expected to develop AKI” is confusing. I assume that this is referring to patients who have a positive BSI compared to those who have a negative BSI? I think it would be more clear to say that you expect that 70% of patients with a positive BSI will go on to develop AKI.

7. PLOS authors have the option to publish the peer review history of their article (what does this mean?). If published, this will include your full peer review and any attached files.

Reviewer #1: No

Reviewer #2: No

---

## [Author Response · Author response to Decision Letter 1]

12 Oct 2020

Reviewer #1: the manuscript provide a valid rationale for the proposed study, with clearly identified and justified research questions. I have no additional comments about this manuscript

Authors’ response 

We thank the reviewer for his comment

Reviewer #2: Thank you for allowing me to review the changes made by Dr. Zaky et. al in their protocol. I commend the authors on their revisions, and I think this version is an improvement. I continue to think the overall aims of the project are valuable, and I have only a few comments that I hope will help with the manuscript.

BSI Components

1. I appreciate the further clarification of the BSI components (10% increase from pre-surgical values in either SWE, Nephrocheck, or NGAL). However, there is no data given to support these cut-points, and it appears to me that this study is expected to generate some of that data. If it is indeed the case that the cutpoints might change based on your findings, it might be useful to explain how you intend to approach creation of the BSI score in the methods, rather than giving specific cut points a priori.

Authors’ response 

We thank the reviewer for his comment. We chose the 10% increase rather than the 0.3 cut off values for cell cycle arrest biomarkers to be consistent with the lack of cut off values for the other components of the BSI. Furthermore, the cut off values of the cell cycle arrest biomarkers are unknown in the preoperative period. Additionally, there is recent evidence to suggest a prediction of AKI at cut off values of less than 0.3 (>0.8 <0.2). We added a paragraph to reflect this justification in the discussion (Pages 17, 18, lines 369-379). We have also alluded to the exploratory nature of this index in the discussion.

2. I know UAB is a leader in NGAL use, but I would be concerned about using that test in this population. NGAL is well known to increase in high-inflammatory states due to production from extra-renal sources. NGAL is likely to go up simply from post-surgical inflammation. I see this was added based on Reviewer #1’s recommendation, but I suspect it will not be a useful addition, personally. I suppose given that this study is designed to optimize the BSI, you could see whether NGAL adds value to the BSI or not.

Authors’ response 

We thank the reviewer for his comment. We included NGAL to explore its role in this population and its trend among open and endovascularly approached aneurysm repairs. as such, the BSI, will be amenable to further refinements in the biomarkers as it is validated in larger group of patients.

3. Nephrocheck has a fairly standard value of 0.3 (as in the PREV-AKI trial, for instance). The 10% change is a metric I’ve not seen before with that marker.

Authors’ response 

We chose the 10% increase rather than the 0.3 cut off values for cell cycle arrest biomarkers to be consistent with the lack of cut off values for the other components of the BSI. Furthermore, the cut off values of the cell cycle arrest biomarkers are unknown in the preoperative period. additionally, there is recent evidence to suggest a prediction of AKI at cut of values of less than 0.3 (>0.8 <0.2). We have added a paragraph to reflect this justification in the discussion (Pages 17, 18, lines 369-379).

4. I continue to be skeptical that there will be significant SWE changes that occur as early as 6 to 24 hours, but I suppose we will find out. I appreciate the inclusion of additional references and discussion.

Sample Size Calculation

5. 40% strikes me as a very high estimate of post-operative AKI, especially given the study design, and doesn’t seem supported by the references provided.

a. Reference 1, Hobson et al.: EVAR: 5.5% – 18%; Branched or fenestrated AAA: 28%; TEVAR: 9.7% (30% after branched aortic dissections); Huber et al. (cited within reference 1) had an AKI rate of 49%, but this seems to be driven by a high number of emergency cases. Emergency cases are an exclusion criteria in this study.

b. Reference 2, Martin-Gonzalez et al. Fenestrated and Branched Endografting: 29% AKI

c. Reference 3, Lee et al; Snorkel approach for EVAR: 32.6% AKI

d. Reference 4, Saratiz et al. Elective EVAR: 18.8% AKI

e. We recently published a study showing a 20.8% AKI rate following elective or urgent aneurysm repair, and several other studies I pulled up in a brief lit search showed AKI rates closer to 15-25% depending on the exact procedure (in elective cases).

My overall concern is that this study will be underpowered with only 80 patients.

Authors’ response 

We thank the reviewer for his comment and agree with his concerns. We based our sample size calculation on several factors. First, our population spans open thoraco-abdominal and abdominal aneurysms repairs known to have a higher incidence of pAKI compared with the endovascular approaches. We have added a new reference (reference 19) that shows the relatively high incidence of pAKI in this population. Second, we have conducted a retrospective analysis of the incidence of pAKI after complex open and endovascular aneurysm repairs at our institution and found an incidence suggestive of 40%. We have made changes in the manuscript (p 6, L 121- L123) to justify our viewpoint.

6. The language regarding “expected to develop AKI” is confusing. I assume that this is referring to patients who have a positive BSI compared to those who have a negative BSI? I think it would be more clear to say that you expect that 70% of patients with a positive BSI will go on to develop AKI.

Authors’ response:

We apologize for the confusion. The change is now made as suggested (P6, L 1211-126).

---

## [Editor Report · Decision Letter 2]

21 Oct 2020

The Bio-sonographic Index. A Novel Modality for Early Detection of Acute Kidney Injury after Complex Vascular Surgery. A Protocol for an Exploratory Prospective Study

PONE-D-20-08584R2

Dear Dr. Zaky,

We’re pleased to inform you that your manuscript has been judged scientifically suitable for publication and will be formally accepted for publication once it meets all outstanding technical requirements.

Kind regards,

Nicholas M Selby, BMedSci BMBS MRCP DM

Academic Editor

PLOS ONE
---

## [Editor Report · Acceptance letter]

4 Nov 2020

PONE-D-20-08584R2 

The Bio-sonographic Index. A Novel Modality for Early Detection of Acute Kidney Injury after Complex Vascular Surgery. A Protocol for an Exploratory Prospective Study 

Dear Dr. Zaky:

I'm pleased to inform you that your manuscript has been deemed suitable for publication in PLOS ONE. Congratulations! Your manuscript is now with our production department. 

Kind regards, 

on behalf of

Professor Nicholas M Selby 

Academic Editor

PLOS ONE